# Implementing High-Flow Nasal Oxygen Therapy in Medical Wards: A Scoping Review to Understand Hospital Protocols and Procedures

**DOI:** 10.3390/ijerph21060705

**Published:** 2024-05-30

**Authors:** Toby Thomas, Yet Hong Khor, Catherine Buchan, Natasha Smallwood

**Affiliations:** 1Melbourne Medical School, University of Melbourne, Corner Grattan Street and Royal Parade, Melbourne 3010, Australia; toby.thomas@swh.net.au; 2Respiratory Research @Alfred, School of Translational Medicine, The Alfred Centre, Monash University, Melbourne 3004, Australia; 3Department of Respiratory and Sleep Medicine, Austin Health, Heidelberg 3084, Australia; 4Institute for Breathing and Sleep, Heidelberg 3084, Australia

**Keywords:** high-flow nasal oxygen, acute respiratory failure, policy, protocol, guideline

## Abstract

Acute hypoxemic respiratory failure (ARF) is a common cause for hospital admission. High-flow nasal oxygen (HFNO) is increasingly used as a first-line treatment for patients with ARF, including in medical wards. Clinical guidance is crucial when providing HFNO, and health services use local health guidance documents (LHGDs) to achieve this. It is unknown what hospital LHGDs recommend regarding ward administration of HFNO. This study examined Australian hospitals’ LHGDs regarding ward-based HFNO administration to determine content that may affect safe delivery. A scoping review was undertaken on 2 May 2022 and updated on 29 January 2024 to identify public hospitals’ LHGDs regarding delivery of HFNO to adults with ARF in medical wards in two Australian states. Data were extracted and analysed regarding HFNO initiation, monitoring, maintenance and weaning, and management of clinical deterioration. Of the twenty-six included LHGDs, five documents referenced Australian Oxygen Guidelines. Twenty LHGDs did not define a threshold level of hypoxaemia where HFNO use was recommended over conventional oxygen therapy. Thirteen did not provide target oxygen saturation ranges whilst utilising HFNO. Recommendations varied regarding maximal levels of inspired oxygen and flow rates in the medical ward. Eight LHGDs did not specify any system to identify and manage deteriorating patients. Five LHGDs did not provide guidance for weaning patients from HFNO. There was substantial variation in the LHGDs regarding HFNO care for adult patients with ARF in Australian hospitals. These findings have implications for the delivery of high-quality, safe clinical care in hospitals.

## 1. Introduction

Acute respiratory failure (ARF) encompasses a range of conditions affecting the respiratory system’s ability to provide oxygen to the body (hypoxemic respiratory failure), or clear carbon dioxide (hypercapnic respiratory failure) [1] and is a common cause for hospital admission in the adult population [2]. The overall disease burden is difficult to assess, but ARF has been associated with a mortality rate of 46% in severe cases [3]. In the general medical ward setting, ARF secondary to coronavirus disease (COVID-19) was associated with a mortality rate of 36% [4]. Determining optimal management for and improving survival from ARF remain key priorities for clinicians.

Therapies for ARF in the ward setting (not including intensive care units (ICUs) or emergency departments) include conventional oxygen therapy (COT), non-invasive ventilation, and, since the mid-2000s, high-flow nasal oxygen therapy (HFNO) (Figure 1) [5]. HFNO is a system that allows the delivery of high flows (up to 60 litres per minute) of humified, heated air together with entrained oxygen [6]. A systematic review by Agarwal et al. found that HFNO reduced the need for invasive ventilation and escalation of therapy compared with COT use among people with COVID-19 and ARF [7]. A separate systematic review by Lee et al. identified that HFNO improved patient comfort, work of breathing, and oxygenation when compared with COT and non-invasive ventilation (NIV) [8].

In 2022, the European Respiratory Society published the first comprehensive clinical practice guideline specifically focused on HFNO, recommending HFNO use as first-line therapy over COT or NIV for patients with severe ARF [9]. Since the COVID-19 pandemic, HFNO has been increasingly utilised in the general medical ward setting for people with ARF to alleviate strain on ICU resources [10,11]. Nevertheless, there is a lack of consensus as to the required resources and safest clinical environment in which to provide HFNO to people with ARF given their risk of clinical deterioration and death.

While the use of evidence based national or international clinical practice guidelines can lead to improved outcomes, the lack of effective implementation, poor adoption, and adherence to guidelines remain major challenges [12]. Healthcare organisations have increasingly attempted to develop and implement local guidelines, policies, procedures, and protocols to complement clinical practice guidelines and drive the implementation of safe, timely, evidence-based healthcare [13]. While local health guidelines are less formalised, they aim to provide information to enable clinicians to make decisions and achieve specific health goals for patients. Guidelines and policies support procedures that instruct how such a policy may be implemented through specific interventions, and protocols provide specific information (e.g., steps of required actions) to operationalise a policy in a standardised way [14]. For task-orientated healthcare interventions such as delivery of HFNO to patients with ARF, local health guidance documents (guidelines, protocols, policies, and procedures) are crucial to ensure safe implementation to the right patient groups [15]. However, it is unclear whether hospitals have specific local health guidance documents (LHGDs) for ward-based administration of HFNO, if these draw on and complement clinical practice guidelines, or what the quality of the information provided in these documents is. Furthermore, variations in LHGDs between institutions may contribute to variations in the quality and safety of patient care.

This study aimed to firstly identify LHGDs regarding the delivery of HFNO to adults with ARF in medical wards in Australian hospitals; secondly, to examine the content and structure of the LHGDs; thirdly, to identify if the advice given in these LHGDs regarding clinical management and HFNO implementation varied between hospitals; and finally, to develop a standardised minimum information set regarding HFNO delivery to adults with ARF in medical wards. We hypothesised that informational content and document structure of LHGDs would vary widely between healthcare organisations.

## 2. Materials and Methods

A scoping review approach was undertaken to identify and examine public hospitals’ LHGDs regarding the delivery of HFNO to adults with ARF in medical wards in the two largest Australian states of Victoria and New South Wales. The search was conducted on 2 May 2022, and then updated on 29 January 2024 using the PROMPT database (https://prompt.org.au/, accessed 29 January 2024). PROMPT is an online management system that houses guidance documents for over 110 public and private health services located in Victoria and New South Wales. This management system supports the sharing of LHGDs amongst participating health services to support the delivery of high quality, safe clinical care in health services, irrespective of their size and resources.

PROMPT was searched using the words ‘nasal high flow oxygen therapy’, which identified any documents that included any one or combinations of these words. This search was supplemented by including LHGDs from Melbourne metropolitan teaching hospitals with which the research team were affiliated and which at the time of the study were not using PROMPT.

### 2.1. Inclusion and Exclusion Criteria

Only LHGDs that met the following inclusion criteria were included in this scoping review: the document was a local guideline, policy, procedure, or protocol;the term ‘nasal high flow oxygen therapy’ or similar was included in the title;the document focused on the provision of HFNO to adult inpatients being managed in medical wards.

Documents were excluded if they were focused on caring for different patient populations (pregnant or paediatric patients) or providing care to people in aged care facilities, at home, or in palliative care settings; focused on the medical treatment of COVID-19 (not delivery of HFNO to people with COVID-19); or described the provision of HFNO to adult inpatients in the emergency department or ICU only. For institutions with multiple documents meeting the inclusion criteria, all qualifying LHGDs were reviewed.

### 2.2. Screening, Data Extraction and Analysis

Title and full text screening was conducted by two independent reviewers (TT and MZ), with any differences resolved via discussion with additional reviewers, NS, YK, and CB. A standardised data extraction tool was developed to extract data regarding the following: 1. Indications for HFNO; 2. HFNO initiation, monitoring, maintenance, and weaning; and 3. Managing clinical deterioration whilst on HFNO and/or any pathway for the escalation of care. Data were extracted and cross-checked independently by two reviewers (TT and MZ), with disagreements resolved by discussion with the full study team. Data were analysed using descriptive statistics using Microsoft Excel (v16.83).

## 3. Results

The initial search yielded 34,352 documents, with 11 additional documents included from two metropolitan Victorian health services not using PROMPT. Following title screening, 32 documents underwent full text screening, with 6 documents then excluded for not meeting eligibility criteria, leaving 26 documents included in the final review (Figure 2). The updated search in 2024 did not identify any new documents for inclusion. Of the 26 included documents from the search on 2 May 2022, 11 were unchanged. Nine documents had been updated with new version numbers or logo changes, but without content changes. Six documents contained minor content changes.

### 3.1. Characteristics of Included Documents

Eight (30.8%) documents were from metropolitan health services, with the remainder (and thus the majority) from regional and rural services. Most (n = 25) LHGDs were from Victorian health services. High-flow nasal oxygen therapy LHGDs were variously described as local guidelines (n = 13, 50.0%), procedures (n = 9, 34.6%), policies (n = 5, 19.2%), and protocols (n = 3, 11.5%). Nineteen documents (73.1%) solely focused on HFNO guidance, with the other seven documents covering HFNO in addition to COT. Three documents (11.5%) additionally contained specific instructions for HFNO use for people with COVID-19 and ARF. The specified target audience for the LHGDs included nursing staff (n = 18, 69.2%), medical staff (n = 17, 65.4%), and physiotherapists, (n = 10, 38.5%), with 5 LHGDs (19.2%) not specifying the target audience.

Reference to national and international clinical practice guidelines regarding acute oxygen therapy or HFNO varied amongst the LHGDs. Five documents referred to the 2015 Thoracic Society of Australia and New Zealand (TSANZ) Acute Oxygen Guidelines [16], three referred to the 2008 British Thoracic Society Guideline for Emergency Oxygen Use in Adults [17]. None referred to the 2017 update [18]. No documents referred to European Respiratory Society guidelines on HFNO [9].

### 3.2. Indications for HFNO

The majority of LHGDs (n = 20, 76.9%) did not provide a definition for hypoxaemia using either oxygen saturation (SpO_2_) values or arterial blood gas (ABG) partial pressure of oxygen (PaO_2_) parameters. When definitions were provided, the SpO_2_ parameter varied (Table 1). All but one document provided examples of indications for HFNO use (i.e., refractory hypoxemia despite conventional oxygen therapy, during breaks from non-invasive ventilation, etc). Most documents (n = 23, 88.5%) listed contraindications for HFNO use. Five (19.2%) LHGDs recommended considering the patient’s goals of care prior to commencement of HFNO. Few documents described the need to communicate HFNO treatment plans with the patient (n = 8, 30.7%) or the patient’s informal carers (n = 4, 15.4%). Seven (26.9%) documents described conditions which may predispose a patient to developing hypercapnic respiratory failure.

### 3.3. HFNO Initiation and Maintenance

Arterial blood gases were rarely recommended prior to initiation of HFNO (Table 2). Most documents did not provide target SpO_2_ ranges to be achieved either at initiation or during ongoing treatment with HFNO. Recommendations regarding patient observations (including type and frequency) required to inform ongoing care and delivery of HFNO varied widely in the LHGDs.

The staff permitted to prescribe or initiate HFNO were not specified in six (23.1%) documents. The seniority of medical, nursing and physiotherapy staff able to initiate HFNO was not specified in 23 (88.5%), 9 (34.6%), and 18 (69.2%) LHGDs, respectively. The clinical discipline (i.e., medical, nursing or physiotherapy) of health professionals who should reassess patients after initiation of HFNO to determine treatment response, ongoing need for HFNO, adverse effects or clinical deterioration was not specified in any of the documents where multiple target audiences were listed (i.e., nurses and medical staff). 

### 3.4. Escalation of HFNO Care

Maximum fraction of inspired oxygen (FiO_2_) and device flow rates that could be safely administered in the medical ward setting were variable. Of 17 documents that specified an upper FiO_2_ limit for the medical ward, the limit varied between 40 and 52%. Similarly, the maximum device flow rate for the medical ward varied between 30 and 60 litres per minute (n = 19) (Appendix A).

Twenty-four (92.3%) LHGDs contained at least one descriptor of when a patient using HFNO should be medically reviewed (e.g., ongoing respiratory distress or persisting hypoxaemia on HFNO). Twenty-one documents (80.8%) described potential adverse events related to HFNO administration. The systems used to detect clinically deteriorating patients varied considerably between hospitals, with the most-used system being the medical emergency team criteria (n = 15, 57.7%), followed by the adult deterioration detection system (n = 6, 23.1%). Eight (30.8%) documents did not specify any system for the escalation of treatment for patients deteriorating whilst using HFNO.

### 3.5. Weaning

Twenty-one documents (80.1%) provided some guidance for weaning patients from HFNO. The time intervals for down-titrating HFNO (e.g., to reduce flow/FiO_2_ every 6 h) were not specified in 22 (84.6%) documents (Table 3). Nineteen documents (73.1%) did not specify what oxygen source (COT, nasal prongs, or face mask) patients should be transitioned to following HFNO. Some documents specified different targets to cease HFNO for patients with and without chronic obstructive pulmonary disease (COPD) (Appendix A).

After assessing the LHGDs and current national and international HFNO recommendations [9,18,19], we developed a HFNO protocol template (Table 4) that could be utilised to standardise the delivery of HFNO to adults with ARF who are being cared for in lower acuity settings such as medical wards.

## 4. Discussion

This is the first scoping review to examine LHGDs regarding ward-based administration of HFNO to adults with ARF. We identified that a small number of health services had specific LHGDs for HFNO use on medical wards, and among those identified, there were several areas of variation or inadequate information, including inconsistent definitions for hypoxemia, variable target saturation ranges for patients being treated with HFNO, limited information regarding detecting deteriorating patients, and minimal guidance regarding safe monitoring of patients during initiation, maintenance, and weaning of HFNO. These inconsistencies are important, as they may contribute to the well-recognised variations in care that occur within countries. Furthermore, inconsistencies in procedural implementation present challenges for clinicians who work across multiple health services, both during routine care but particularly during the COVID-19 pandemic when HFNO was used widely on medical wards [20].

While LHGD variance may be attributed to a lack of high-quality evidence to support policy development [21], it is worth noting that most of the LHGDs identified in this study did not refer to national or international acute oxygen or HFNO clinical practice guidelines, and even recommendations supported by strong evidence, such as the need to maintain patient saturations within a target range [19], were not found consistently. The European Respiratory Society Guidelines for HFNO [9] summarise the recent evidence regarding HFNO and provide key indications for when HFNO should be utilised compared with other oxygen or ventilatory support interfaces in different acute settings. At present, the LHGDs reviewed do not consistently reflect ERS clinical practice guidelines recommendations. While many of the LHGDs predated the ERS guidelines, updates to LHGDs that reflect current international guidelines are required and may support the standardisation of care.

Another option to address the issue of LHGD inconsistency is through the development, testing, and implementation of standardised templates for certain procedures (such as initiation of HFNO or NIV). The template could then be customised to meet the specific needs of individual health services.

Although critical for assessing respiratory status and informing clinical decisions, most LHGDs did not recommend an oxygen saturation (SpO_2_) or ABG value to define hypoxemia and thus guide clinicians as to when to consider initiating HFNO. The Thoracic Society of Australia and New Zealand Acute Oxygen Guidelines recommend initiating COT when people with ARF have SpO_2_ < 92% (or <88% for people at risk of hypercapnia) [19]. Both parameters are also suitable when considering commencing HFNO. Furthermore, we found that less than half of the LHGDs provided target oxygen saturation ranges for patients receiving HFNO. The clinical importance of oxygen titration is well documented; a meta-analysis by Chu et al. [22] revealed that for every 1% rise in SpO_2_, the relative risk of in-hospital death for acutely unwell adult inpatients rose by 25%. Adverse effects of hyperoxia are particularly relevant for patients with COPD, where one study found patients who receive titrated oxygen therapy during acute exacerbation have a 58% reduced risk of death [23]. 

A significant concern when providing HFNO outside of high acuity wards (such as ICU or HDU) is that deteriorating patients may not be recognised promptly, and thus not receive timely escalation of care. Patients with ARF on the ward are likely to require an ‘escalation therapeutic strategy’ [24] to avoid fatal complications. The importance of prompt recognition of patient deterioration is emphasised by the increased morbidity and mortality observed in deteriorating HFNO patients who are intubated after 48 h [25]. We found that a significant portion of the LHGDs did not specify an alert system for staff to promptly detect patient deterioration and initiate rapid escalation of care [26], which is crucial for safe delivery of HFNO to patients with ARF outside of high acuity wards. Similarly, physiological parameters such as respiratory rate, work of breathing and SpO_2_ are important predictors of patient deterioration and have been incorporated into clinical assessment tools like the ROX index, which predicts the likelihood of HFNO failure [27,28]. We found that instruction to document these parameters varied across the LHGDs and were often omitted. The need for higher HFNO flow rates or FiO_2_ has also been cited as a useful trigger to recognise patients who require escalation of care such as ICU admission [19]. Very few LHGDs in this study documented upper limits for HFNO flow rate or FiO_2_ for patients being cared for in the general medical ward setting.

While this study has identified variation in HFNO LHGDs, future research should seek to clarify several practical aspects of LHGD use. It remains uncertain whether clinicians are aware of, and actively use these documents in routine clinical practice. A previous study on utilisation of hospital policies by nursing staff found that local policies were often under-utilised [29]. Similarly, a study by Cousins et al. [30] found that 37% of surveyed Australian practitioners were not aware of the TSANZ Acute Oxygen Clinical Practice Guidelines [16]. Future research could investigate the extent to which clinicians use LHGDs and explore the factors that facilitate or hinder their implementation. Furthermore, studying patient outcomes in health services with comprehensive HFNO LHGDs could provide insights into the importance of high-quality LHGDs for delivering optimal HFNO care.

### Strengths and Limitations

To our knowledge, this is the first study of its kind to assess LHGDs to determine if recommendations are evidence based, consistent, and sufficiently detailed to support clinical implementation. Our study adopted a scoping review approach with a defined protocol and rigorous methodology. We utilised the PROMPT LHGD sharing platform, which is used by over one hundred hospitals in the two largest Australian states (which collectively have a population of over 14 million people [31]), and included LHGDs from a mix of metropolitan, rural, and regional hospitals. This study has a limitation that warrants discussion. It could be postulated that as not all health services use the PROMPT platform, the results may not be a true representation of all the guidance documents used for HFNO in the ward setting. However, we were also able to include LHGDs from some hospitals that did not use the PROMPT platform. Despite these being a small proportion of the documents, it is likely that inclusion of a higher number of these documents in this study would not change the results significantly. Even considering this small limitation, this is a first-of-its-kind study and is important in highlighting variations in HFNO local health documents that have not been identified previously.

## 5. Conclusions

This innovative study has demonstrated significant variation in the structure and content of HFNO local health guidance documents for adult patients with acute respiratory failure in Australian hospitals. Most notably, we identified major variations in guidance regarding the initiation and titration of HFNO, as well as recognising patient deterioration whilst using HFNO. These findings have implications for the delivery of high-quality, evidence-based, safe clinical care when using HFNO in the medical ward setting. Future research should aim to consider how to standardise LHGDs to minimise variations in care and to understand how clinicians currently use LHGDs in routine practice. It may also assess the variance across international HFNO LHGDs and their alignment with established clinical practice guidelines.

## Figures and Tables

**Figure 1 ijerph-21-00705-f001:**
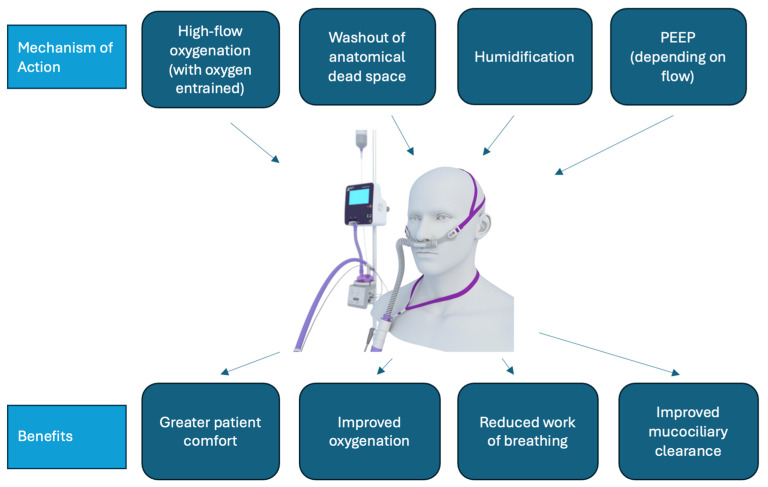
HFNO mechanism of action and benefit. PEEP: Positive end-expiratory pressure.

**Figure 2 ijerph-21-00705-f002:**
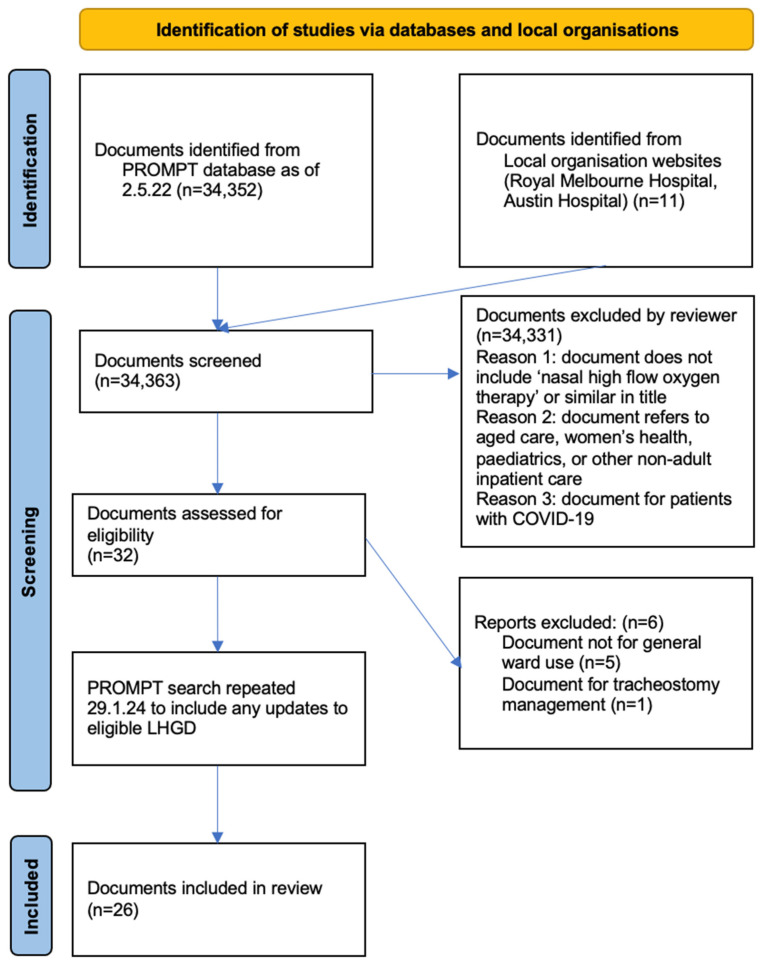
PRISMA search strategy.

**Table 1 ijerph-21-00705-t001:** Indications for HFNO.

	Count (n)	Frequency (%)
SpO_2_ hypoxemia definition to trigger HFNO consideration		
-Definition not specified	20	76.9
-<86%	1	7.7
-<90%	2	7.7
-<92–93%	2	3.8
-<95%	1	3.8
PaO_2_ (ABG) hypoxemia definition to trigger HFNO		
-Definition not specified	24	92.3
-<65 mmHg	2	7.7
Hypoxemia assessed with *		
-Not specified	2	7.7
-Pulse oximetry	23	88.5
-Arterial blood gas	12	46.2
Minimum oxygen flow rate given before initiation of HFNO		
-Not specified	15	57.7
-1–6 LPM	2	7.7
->4 LPM	5	19.2
->6 LPM	4	15.4
Minimum conventional oxygen therapy FiO_2_ given before initiation of HFNO		
-Not specified	23	88.5
->40%	3	11.5
SpO_2_ targets (lower level of range) once HFNO initiated		
-Not specified	13	50.0
-92%+	6	23.1
-93%+	7	26.9
SpO_2_ targets (upper level of range) once HFNO initiated		
-Not specified	18	69.2
-96%	7	26.9
-98%	1	3.8

* Categories are not mutually exclusive and cumulative percentages may sum to over 100. SpO_2_: Pulse oximetry oxygen saturation, HFNO: High-flow nasal oxygen, PaO_2_: Partial pressure of oxygen, ABG: Arterial blood gas, LPM: Litres per minute, FiO_2_: Fraction of inspired oxygen.

**Table 2 ijerph-21-00705-t002:** Recommendations for initiation and maintenance of HFNO.

	At Set Up of HFNO		After Initiation of HFNO	
	Count (n)	Frequency (%)	Count (n)	Frequency (%)
ABG recommended	4	15.4	N/A	N/A
VBG used to assess need for HFNO	2	7.7	N/A	N/A
Additional SpO_2_ target range provided for patients at risk of hypercapnic respiratory failure	10	38.5	N/A	N/A
HFNO device flow rate			N/A	N/A
-Not specified	8	30.8		
-To meet patient requirements	14	53.8		
-30 LPM	4	15.4		
Initial FiO_2_			N/A	N/A
-Not specified	9	34.6		
-To meet patient requirements	16	61.5		
-50%	1	3.8		
Documentation of HFNO settings *	N/A	N/A		
-Device flow rate			16	61.5
-FiO_2_			15	57.7
-Delivered air temperature			9	34.6
-Target SpO_2_ range			2	7.7
Documentation of patient observations *	N/A	N/A		
-SpO_2_			22	84.6
-Respiratory rate			19	73.1
-Work of breathing			15	57.7
-Heart rate			9	34.6
-Blood pressure			9	34.6
-Conscious state			7	26.9
Time until first observations			N/A	N/A
-Not specified	9	34.6		
-0–1 h	13	50.0		
-1–4 h	4	15.4		
Observation frequency	N/A	N/A		
-Not specified			5	19.2
-Clinical discretion			5	19.2
-0–1 h			6	23.1
-1–4 h			9	34.6
-4+ h			1	3.8
Diagram provided for HFNO set up/equipment	24	92.3		

* Categories are not mutually exclusive and cumulative percentages may sum to over 100. SpO_2_: Pulse oximetry oxygen saturation, HFNO: High-flow nasal oxygen, ABG: Arterial blood gas, VBG: Venous blood gas, LPM: Litres per minute, FiO_2_: Fraction of inspired oxygen. N/A: Not available.

**Table 3 ijerph-21-00705-t003:** Recommendations for weaning HFNO.

Characteristic	Count (n)	Frequency (%)
Weaning to maintain target SpO_2_ range	11	42.3
Patient observations to be documented *		
-Not specified	10	38.5
-Respiratory rate	14	53.8
-Work of breathing	14	53.8
-SpO_2_	13	50.0
-Heart rate	7	26.9
-Blood pressure	7	26.9
-Consciousness	5	19.2
FiO_2_ targets before HFNO cessation **		
-Not specified	20	76.9
-FiO_2_ < 40%	2	7.7
-FiO_2_ < 30%	2	7.7
-FiO_2_ 21%	2	7.7
Flow rate targets before HFNO cessation **		
-Not specified	15	57.7
-Device flow < 30 LPM	10	38.5
-Device flow < 10 LPM	1	3.8

* Categories are not mutually exclusive and cumulative percentages may sum to over 100. ** Patients without chronic obstructive pulmonary disease (see Appendix A). SpO_2_: Pulse oximetry oxygen saturation, HFNO: High-flow nasal oxygen, LPM: Litres per minute, FiO_2_: Fraction of inspired oxygen.

**Table 4 ijerph-21-00705-t004:** HFNO LHGDs template.

Document Purpose	To Provide Clinical Guidance for Initiating and Managing HFNO Therapy to Adults with ARF in the Ward Setting
Target audience	Medical, nursing, and physiotherapy staff trained in HFNO use for adult patients.
Definitions	Hypoxaemia requiring HFNO use: ABG PaO_2_ < 60 mmHg or SpO_2_ < 92% Note: Target SpO_2_ range may vary depending according to individual patient factors e.g., SpO_2_ < 88% in the setting of hypercapnia.
Indications	First-line oxygen therapy for patients with acute hypoxemic respiratory failure: patients not at risk of hypercapnic respiratory failure: SpO_2_ < 92%; patients at risk of hypercapnic respiratory failure (e.g., those with chronic obstructive pulmonary disease): SpO_2_ < 88%. Intermittent use during breaks from non-invasive ventilation. Tracheostomy patients requiring humidified air. As a palliative measure to manage dyspnoea as part of end-of-life care (please refer to ‘HFNO in ward-based palliation’). The goals of care and patient preferences must be considered and documented for each individual.
Contraindications	Recent nasal surgery Recent maxillofacial trauma Suspected base of skull fracture Raised intracranial pressure Persistent epistaxis Pneumothorax Increased work of breathing without hypoxemia (except as a palliative measure)
Initiation	As SpO_2_ and venous blood gases have significant limitations, ABG measurement should be considered when clinically appropriate, particularly when hypercapnic respiratory failure is considered. Steps to set up HFNO equipment (local service to add information). Titrate HFNO device parameters (device flow rate and FiO_2_) to maintain oxygen saturation in target range: Patients not at risk of hypercapnic respiratory failure: SpO_2_ 92–96%; Patients at risk of hypercapnic respiratory failure: SpO_2_ 88–92% *. * Conditions which increase the risk of hypercapnic respiratory failure, e.g., severe chronic obstructive pulmonary disease, respiratory muscle weakness, severe kyphoscoliosis, or a history of hypercapnia or respiratory acidosis. Regularly monitor and document the response to HFNO, including measuring the following parameters (ideally every 15 min in the first hour of therapy, although likely to be influenced by nursing ratios and clinical circumstance): Respiratory rate; Pulse oximetry; Heart rate; Temperature. Adjust and document HFNO device parameters to maintain oxygen saturation in target ranges (as stated above): FiO_2_ (by the oxygen flow rate); Device flow rate (LPM).
Maintenance	Once stable, observation frequency will vary as clinically indicated (i.e., from one to four hourly). Documentation of patient parameters should include: Respiratory rate; Pulse oximetry; Heart rate; Temperature. Documentation of machine parameters should include: FiO_2_ (by the oxygen flow rate); Device flow rate (LPM). ABG may be considered as per clinical need on an ongoing basis.
Criteria for medical review	Increasing Respiratory rate Oxygen requirements (FiO_2_) Device flow rates (LPM) If patient is unable to meet SpO_2_ target range Others as indicated by the standard local hospital alert system
Escalation pathway	Patients with ARF using HFNO on a medical ward may deteriorate, therefore require regular clinical assessment and monitoring. Use local documented criteria to escalate care for a deteriorating patient: e.g., medical emergency team criteria, early warning score or other deterioration detection system, local ward FiO_2_/flow limits Referral pathway to ICU (include contact method e.g., ICU Registrar Pager)
Weaning	Aim to maintain patient oxygen saturations in the target range. Wean FiO_2_ < 40%. Wean flow to <30 LPM. On cessation of HFNO, transition the patient to conventional oxygen therapy as required. Once HFNO ceased, monitor patient parameters 4–6 times hourly (or more frequently if clinically required): Respiratory rate; Pulse oximetry; Heart rate; Temperature.
HFNO in ward-based palliation	HFNO may be used on the ward to manage dyspnoea at end of life, as it is often well tolerated and does not interfere with a patient’s ability to communicate. HFNO provided for symptom palliation does not usually require the monitoring or escalation criteria described above. Treatment should be adjusted to patient comfort.
Associated documents	e.g., Oxygen therapy LHGDs, escalation of care LHGDs, HFNO user manual
References	e.g., ERS HFNO clinical practice guidelines [9], TSANZ Acute Oxygen Guidelines [19].
Authorship and review date	Contributing staff members and positions Date of last LHGDs review Date LHGDs due for review

SpO_2_: Pulse oximetry oxygen saturation, HFNO: High-flow nasal oxygen, PaO_2_: Partial pressure of oxygen, ABG: Arterial blood gas, LPM: Litres per minute, FiO_2_: Fraction of inspired oxygen.

## Data Availability

The data used are not publicly available to maintain the anonymity of the health services that were included in the study.

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
