# Peer review of "Implementing High-Flow Nasal Oxygen Therapy in Medical Wards: A Scoping Review to Understand Hospital Protocols and Procedures"

_ijerph, 2024, doi:10.3390/ijerph21060705_

Round 1
Reviewer 1 Report
Comments and Suggestions for Authors
This scoping review is very well-written and useful in starting to standardise the LHGDs regarding High-flow nasal cannula (HFNC) oxygen therapy, which is a relatively new modality and is gaining traction globally as a treatment for respiratory failure. I would like to make the following recommendations:
1) To replace HFNO to HFNC (High Flow Nasal Cannula) as this is the more internationally recognised term and definitely the preferred one in Europe. This is mainly because HFNO can be ambiguous as high flow oxygen has hitherto implied high FiO2
2) In clinical practice, many patients receive ward-based HFNC for the comfort associated with relief to work of breathing and mucociliary clearance facilitated by heated humidification. Therefore the scoping review will be more useful if the suggested LHGD template (Table 4) includes a section on use of ward-based HFNC in patients being treated with palliative intent (the monitoring and escalation sections imply all patients receiving HFNC are for escalation to critical care (high acuity wards).
Reviewer 2 Report
Comments and Suggestions for Authors
I read this paper with great interest. The paper addresses a highly relevant issue and search strategy and data extraction appear to be thoroughly conducted, involving multiple databases and direct hospital contacts, ensuring a wide range of documents were reviewed, which enhances the robustness of the findings. English language is fine.
I have some major concerns before ensuring publication:
1. The study is geographically limited to Australian hospitals, which might limit the applicability of the findings globally. Expanding the research to include international guidelines could enhance the relevance and applicability of the research outcomes.
2. The paper effectively identifies inconsistencies in the guidelines regarding hypoxaemia definitions, monitoring, and escalation protocols. IMHO, authors should perform a deeper analysis of why these inconsistencies exist and how they affect patient outcomes.
3. The paper lacks a critical discussion of the latest international research and guidelines, such as those from the European Respiratory Society, which have been pivotal in HFNO therapy. Integration and discussion of these guidelines would strengthen the argument for standardization.
4. An eye-catching figure describing how HFNO works would be nice for readers.
5. I found a couple of typos throughout the paper. Please recheck.
Round 2
Reviewer 2 Report
Comments and Suggestions for Authors
authors replied to my comments in a satisfactorily way. Therefore, IMHO this paper can now be accepted.